# Advancements in SiC-Reinforced Metal Matrix Composites for High-Performance Electronic Packaging: A Review of Thermo-Mechanical Properties and Future Trends

**DOI:** 10.3390/mi14081491

**Published:** 2023-07-25

**Authors:** Liyan Lai, Bing Niu, Yuxiao Bi, Yigui Li, Zhuoqing Yang

**Affiliations:** 1School of Science, Shanghai Institute of Technology, Shanghai 201418, China; bingniu1552@163.com (B.N.); biyuxiao1999@163.com (Y.B.); ygli@sit.edu.cn (Y.L.); 2National Key Laboratory of Science and Technology on Micro/Nano Fabrication, School of Electronic Information and Electrical Engineering, Shanghai Jiao Tong University, Shanghai 200240, China; yzhuoqing@sjtu.edu.cn

**Keywords:** electronic packaging, metal matrix composites, silicon carbide, thermo-mechanical properties

## Abstract

With the advancement of semiconductor technology, chip cooling has become a major obstacle to enhancing the capabilities of power electronic systems. Traditional electronic packaging materials are no longer able to meet the heat dissipation requirements of high-performance chips. High thermal conductivity (TC), low coefficient of thermal expansion (CTE), good mechanical properties, and a rich foundation in microfabrication techniques are the fundamental requirements for the next generation of electronic packaging materials. Currently, metal matrix composites (MMCs) composed of high TC matrix metals and reinforcing phase materials have become the mainstream direction for the development and application of high-performance packaging materials. Silicon carbide (SiC) is the optimal choice for the reinforcing phase due to its high TC, low CTE, and high hardness. This paper reviews the research status of SiC-reinforced aluminum (Al) and copper (Cu) electronic packaging materials, along with the factors influencing their thermo-mechanical properties and improvement measures. Finally, the current research status and limitations of conventional manufacturing methods for SiC-reinforced MMCs are summarized, and an outlook on the future development trends of electronic packaging materials is provided.

## 1. Introduction

With the continuous advancement of microelectronics technology, the power load capacity of electronic devices and systems is constantly increasing, leading to a continuous rise in heat generation [1,2]. At the same time, the continuous improvement in circuit integration further increases the heat generation intensity of chips per unit area, making cooling technology one of the main challenges that need to be overcome for the further development of high-performance chips. For example, power chips widely used in high-tech equipment such as 5G base stations, high-voltage substations, high-speed trains, and radars exhibit heat generation intensities significantly exceeding 1 kW/cm^2^, reaching the range of ultra-high heat flux density. However, due to the lack of micro-cooling technology that matches these heat levels, the working temperature of the chips increases, leading to shortened operational lifespan, decreased capabilities, and reduced reliability. This forms a practical “thermal constraint” that hinders further development. In reality, the majority of failures in high-power electronic systems are caused by excessively high working temperatures. Research has shown that for every 10 °C increase in working temperature, the product’s lifespan is reduced by half [3].

Currently, research aimed at enhancing chip cooling capacity primarily focuses on two approaches. The first approach involves the development of efficient micro-scale heat dissipation mechanisms, while the second approach focuses on the development of high-performance thermal management materials. In the first approach, researchers are dedicated to designing packaging substrates, also known as heat sinks, with high cooling capabilities. Currently, the highly efficient liquid cooling technology, particularly microchannel liquid cooling technology, is receiving significant attention. In the second approach, researchers are focused on the development of packaging materials with excellent thermal properties, also known as thermal management materials. The research primarily revolves around high-performance thermal sink materials, especially those with high thermal conductivity (TC) and low coefficient of thermal expansion (CTE) [3,4,5,6]. These materials are widely used in the production of chip packaging substrates, which are currently the most common method of chip heat dissipation and serve as the material foundation for other novel and efficient heat dissipation mechanisms and devices.

As an ideal thermal management material, it should possess at least the following four characteristics [7]: High TC to reduce thermal resistance and enhance the cooling capacity of the thermal sink; Low CTE to minimize thermal mismatch and improve thermal cycle stability; Excellent micro-processing and shaping potential to meet the requirements of precision microforming manufacturing, laying the foundation for the physical implementation of efficient heat exchange mechanisms; Good mechanical strength to maintain the stability of fine microstructures under higher operating pressures, meeting the basic requirements for stable operation of efficient heat exchange mechanisms. In addition to these aspects, factors such as low cost, availability, and environmental stability are also important conditions for widespread application and should be appropriately considered.

Compared to the standards of ideal thermal management materials, metal matrix composites (MMCs) consisting of a high TC matrix metal and reinforcing phases have the most advantageous development potential. They are one of the important research directions in the field of electronic packaging materials [8]. This paper focuses on the relevant research work regarding the application of silicon carbide (SiC) reinforced MMCs in electronic packaging. It briefly outlines the classification of electronic packaging materials and the current research status of metal-based electronic packaging materials. The discussion and review primarily focus on the factors influencing the properties of SiC-reinforced aluminum(Al)-based and copper(Cu)-based composites. Furthermore, it provides a reference for the existing issues and future development directions of thermal management materials.

## 2. Classification of Electronic Packaging Materials

Electronic packaging materials can be classified into ceramic packaging materials, plastic packaging materials, and metal packaging materials based on their material types [9].

Ceramic packaging materials have lower CTE and density, but their TC is relatively low, which hinders the timely dissipation of heat generated by the chips [10,11]. Plastic packaging materials offer advantages such as small size, light weight, and high impact resistance. However, internal defects like voids can reduce their TC, and they are susceptible to corrosion and damage in harsh environments [12,13]. Metal materials exhibit excellent TC, but their CTE differs significantly from that of chip materials. The thermal cycling during the operation of electronic devices generates large internal stresses, posing risks of damage or failure to the devices [14,15].

With the rapid development of electronic information technology, traditional electronic packaging materials have fallen behind the pace of microelectronics technology. The increasing demands of microelectronic and semiconductor devices for packaging materials have accelerated the development of advanced MMCs [16].

## 3. Current Status of Metal-Based Electronic Packaging Materials

MMCs are advanced materials formed by combining metals or alloys as matrices with reinforcements such as fibers, whiskers, or particles through composite fabrication techniques [17]. Numerous studies have demonstrated that the comprehensive properties of composites surpass those of the base matrix materials [18,19,20,21,22], thereby meeting various requirements. Particulate and whisker-reinforced non-continuous MMCs offer cost-effectiveness and process ability [23]. The incorporation of non-continuous reinforcements significantly improves the wear resistance, thermal and mechanical properties, elastic modulus, and reduces CTE of metals [24].

The addition of particulate reinforcements enhances the strength, wear resistance, and tensile strength of MMCs, but the reinforcing effect is not as significant as that of whiskers [25]. Whiskers, as reinforcements, can enhance the strength, wear resistance, tensile strength, corrosion resistance, thermal stability, and TC of metal materials [26]. Commonly used whiskers include silicon carbide whiskers (SiCw) [27], aluminum oxide whiskers (Al_2_O_3_w) [28], calcium carbonate whiskers (CaCO_3_w) [29], silicon nitride whiskers (Si_3_N_4_w) [30], and zinc oxide whiskers (ZnOw) [31]. Among these, SiCw is most commonly used due to its highest hardness, maximum modulus, greatest tensile strength, and highest heat resistance among the synthesized whisker products. It is available in two forms: α-type and β-type, with the latter exhibiting superior overall properties compared to the former [32,33]. SiCw is widely used as reinforcement. Xu Zhang et al. [34] prepared SiCw/Al composites by hot isostatic pressing process. Tayebi M et al. [35] prepared SiCw/Mg composites by stir casting. Jiang Feng et al. [36] prepared SiCw/Cu composites by powder metallurgy and hot extrusion. Liyan Lai et al. [37,38] prepared SiCw/Ni composites with excellent mechanical properties by electrodeposition.

Currently, particles are mostly used as reinforcements to enhance metal materials. silicon carbide particle (SiCp)-reinforced MMCs exhibit excellent electrical properties and TC, as well as outstanding mechanical properties [39]. Aluminium oxide particle (Al_2_O_3_p)-reinforced Al-based composites demonstrate significant improvements in hardness and tensile strength [40]. Silicon nitride particle (Si_3_N_4_p)-reinforced Al-based composites can achieve compressive super-plasticity under certain processing conditions [41]. Titanium carbide particle (TiCp)-reinforced Cu-based composites exhibit increased hardness and density, as well as reduced friction coefficient [42]. Boron carbide particle (B_4_Cp)-reinforced Al-based composites possess high strength and elastic modulus [43].

MMCs can be classified into Al, magnesium (Mg), Cu, Zn, and nickel(Ni)-based composites based on the matrix type. Al and Cu-based MMCs exhibit excellent thermal and mechanical properties, making them widely used in electronic packaging [44,45]. The reinforcement materials mainly include Al_2_O_3_, SiC, AlN, among which SiC is extensively utilized due to its CTE (approximately 4 × 10^−6^ K^−1^) and relatively lower price [44,45,46,47,48,49].

### 3.1. SiC/Al Composite Materials

SiC/Al composite materials exhibit remarkable properties, including high specific strength, high specific modulus, high hardness, wear resistance, good thermal stability, and high fatigue strength. They can be manufactured and processed using conventional methods, making them highly promising for electronic packaging materials [50,51].

Due to the relatively high CTE of Al alloys, methods such as increasing SiC volume fraction are employed to match their CTE with semiconductor materials (Si, Germanium (Ge)). As SiC volume fraction increases, the residual stress in the composite material becomes larger, significantly influencing its properties. Additionally, byproducts generated at high temperatures can also affect TC of the composite material. Therefore, this subsection primarily discusses the research status of SiC/Al composite materials from the perspectives of preparation methods, residual stress, and interfacial reactions.

#### 3.1.1. Preparation Methods

Due to the limitations of cost and process, the traditional powder metallurgy method and stir casting method are difficult to be applied to the manufacture of high volume fraction SiC/Al composites [52]. Low-pressure infiltration in an ambient atmosphere offers advantages such as low cost and simplicity, making it a currently cost-effective method for the industrial-scale production of SiC/Al composites [53].

Cui et al. [54] utilized the low-pressure infiltration method to prepare composites with a SiC volume fraction of 60%, density of 2.94 g/cm^3^, elastic modulus of 220 GPa, TC of 180 W/(m·K), and CTE of 8.0 × 10^−6^ K^−1^. Huang [55] also employed the low-pressure infiltration method to produce SiCp/Al composites with SiC particle volume fractions ranging from 50% to 70%. These composites exhibited a high TC of 170 W/(m·K) and a CTE ranging from 6.6 × 10^−6^ to 9.7 × 10^−6^ K^−1^, meeting the thermal physical property requirements for packaging materials.

The squeeze casting method offers a simple process and produces composites with fine and stable structures. Zhang et al. [56] proposed squeeze casting to fabricate SiCp/Al composites with high reinforcement content for electronic packaging applications. These composites had SiC particle volume fractions ranging from 50% to 70%, CTE ranging from 8.3 × 10^−6^ to 10.8 × 10^−6^ K^−1^, and flexural strengths exceeding 370 MPa. Lee et al. [57] also utilized squeeze casting to fabricate SiCp/Al composites with a low CTE (8.0 × 10^−6^ K^−1^) and high TC (155 W/(m·K)).

Wang Wujie [58] prepared SiCp/6061Al composites with a SiC volume fraction of 50% by high-speed pressing plus atmospheric pressure sintering and discharge plasma sintering, respectively, and studied the influence of pressing height on the thermal conductivity of the composites and the influence of discharge plasma sintering pressure on the thermal conductivity of the composites in the former method. The change of pressing height to the thermal conductivity of the composite is shown in Table 1, and the main reason for the decrease of the thermal conductivity of the composite with a pressing height of more than 60 cm is that the integrity of SiCp in the compacted is gradually destroyed, which reduces the thermal conductivity of the composite. The thermal conductivity of the composite under different sintering pressures is shown in Figure 1, and the thermal conductivity of the composites gradually improves as the sintering pressure increases.

Powder metallurgy possesses advantages in preparing composites with high volume fractions. However, conventionally prepared powder metallurgy composites have lower density and are limited in terms of size and shape. Improved pseudo-semi-solid thixotropic forming methods have demonstrated better density and reinforcement volume fractions compared to powder metallurgy [59].

Guo et al. [60] employed a pseudo-semi-solid thixotropic forming process to prepare SiCp/Al electronic packaging materials with three different volume fractions: 40%, 56%, and 63%. They measured the density, compact density, TC, CTE, compressive strength, and flexural strength of the composite materials as a function of the reinforcement volume fraction. The experimental results, as presented in Table 2, demonstrate the good thermal stability of SiCp/Al composites and their CTE matching with Si, the chip material. The performance, as illustrated in Figure 2, meets the requirements of electronic packaging materials.

#### 3.1.2. Residual Stress

The residual stress has many adverse effects on the comprehensive performance of the composite material, and the research on the residual stress of SiCp/Al composite material has been deepened by relevant scholars at home and abroad [61,62,63]. The causes of residual stress in the high-temperature preparation process of SiC/Al composites are mainly as follows: 1. residual stress caused by temperature gradient; 2. residual stress caused by the difference in thermal expansion coefficient of matrix and reinforcement; 3. residual stress caused by local volume change caused by interface reaction or matrix microplastic deformation [64].

Al has a CTE of approximately 23 × 10^−6^ K^−1^, while SiC has a CTE of approximately 4 × 10^−6^ K^−1^. Due to the significant difference in CTE between the metal matrix Al and the ceramic reinforcement SiC, residual stresses are generated during the preparation process of SiC/Al composites [65,66,67]. Furthermore, as the volume fraction of SiC increases, the residual stress in the composite material becomes greater, leading to adverse effects on its mechanical and physical properties [14].

Mou et al. [68] performed stress analysis on SiCw/Al composites prepared by die casting and proposed the concept of a zero residual stress temperature (Tc). They derived an expression for Tc using micromechanical models and elastic-plastic theory. The results showed that Tc is related to the properties of the constituent materials, the difference in CTE, the degree of processing hardening of the matrix, temperature differences, and the volume fraction of whiskers. The experimental and predicted results were consistent, providing a theoretical basis for reducing residual stresses during the processing of composite materials.

According to the research results at this stage, aging treatment, cold and heat cycle treatment and other heat treatment methods have certain effects on eliminating or reducing the thermal residual stress in SiC/Al composites [69]. KIM et al. [70] investigated the effects of different heat treatment processes, such as water quenching, air cooling, and furnace cooling, on CTE of SiCp/Al composites fabricated by squeeze casting. They observed a significant reduction in CTE of high-volume fraction SiCp/Al composites when subjected to heat treatment, especially water quenching.

Li Jing et al. [71] carried out solution aging treatment on SiC/Al composites at different temperatures and tested the residual stress on the surface of the composites under different solution temperatures and aging states. The residual stress of composite materials at different solution temperatures is shown in Figure 3. With the increase of solution temperature, the residual stress of composite materials after solution aging treatment firstly increases and then decreases. The analysis shows that the residual stress mainly comes from two aspects: First, the lattice distortion caused by GP zone [72] of continuous phase solid solution aging on aluminum base generates stress field; Second, in the process of solution aging heat treatment, the thermal expansion coefficient between SiC and aluminum is very different, resulting in a large residual stress between the interface.

In addition to the full study of the residual stress level of SiCp/Al composites in the experimental aspect, in the finite element simulation, it is found that the distance, volume fraction, shape and interaction between SiC particles can cause the change of residual stress in SiC/Al composites [73,74].

The thermal residual stresses resulting from temperature variations usually distribute near the interfaces between the reinforcement and the matrix particles [69]. S. Ho and Saigal [75] conducted finite element analysis on the thermal residual stresses in SiCp/Al composites. The results indicated that the highest residual stress occurred at the interface between SiC particles and Al, and the residual stress decreased as the distance from the interface increased.

The shape, size, and distribution of the reinforcement particles also influence the residual stresses in the composite material. Han et al. [76] performed finite element analysis on the thermal stresses of Al-based composites reinforced with SiC particles of different shapes. The experimental results showed that for nearly spherical SiC particles, the thermal residual stresses near SiC/Al interface were evenly distributed, and the residual stresses decreased with increasing distance from the interface. In the case of angular SiC particles, the thermal residual stresses near the SiC/Al interface were unevenly distributed, with stress concentrations and plastic strain concentrations occurring at the corners of SiC particles in Al matrix.

To mitigate the adverse effects of residual stresses on the properties of composite materials, Shen et al. [77] conducted finite element simulations and found that the preparation of composites with brittle phase reinforcement and ductile matrix could lead to increased CTE due to the development of residual stresses. Post-heat treatment can be employed to eliminate the influence of residual stresses on the physical properties of materials.

Liu et al. [78] used COMSOL finite element software to construct a two-dimensional random distribution model of SiC/Al composites, and studied the influence of the addition of the second reinforced phase graphene (Gr) [79] on the thermal residual stress of SiC/Al composites. The results show that the average residual stress of composites decreases slowly with the increase of Gr volume fraction. After adding 5 vol%Gr to SiC/Al composites with 40% SiC volume fraction, the average residual stress of composites decreases by 82 MPa. The main reason for the decrease of residual stress is that the weak interface binding energy of Gr and aluminum remakes partial thermal stress. The average residual stress of the composite is reduced.

#### 3.1.3. Interface Reactions

The properties of SiC/Al composites are closely related to the microstructure of the interfacial phase formed in the preparation process and the substances generated by the interfacial reaction [80,81,82]. Slight interface reaction can improve the infiltration and combination of matrix and reinforcement, which is beneficial to the properties of composite materials, but severe interface reaction will form brittle interface phase, which seriously affects the properties of composite materials [83]. 

Adding appropriate amount of magnesium into the aluminum matrix, Silicon dioxide (SiO_2_) on the surface of SiCp can have a slight interfacial reaction with magnesium, forming MgAl_2_O_4_ between the interfacial phases, improving the wetness of the interface phase of the composite, which is conducive to improving the performance of the composite. In the process of SiC/Al composites prepared at high temperature and pressure, serious interfacial reactions will occur, and by-product aluminium carbide (Al_4_C_3_) will be generated, and Al_4_C_3_ will become the thermal diffusion barrier layer at the interface, which will greatly reduce the thermal conductivity of the composite [84,85].

Cui et al. [86] prepared SiCp/Al composites with a volume fraction of 55 vol% by non-pressure infiltration method, and studied the spatial distribution of harmful interfacial product Al_4_C_3_ near the SiC/Al infiltration interface of the composites. The results showed that the interfacial product Al_4_C_3_ gradually decreased with the increase of infiltration depth, that is, the interfacial reaction degree decreased along the infiltration direction.

Currently, there are two main approaches to suppress interface reactions in SiC/Al composites. The first approach involves surface treatment of SiC, while the second approach involves the addition of alloying elements to Al matrix. The addition of alloying elements to Al not only inhibits interface reactions but also enhances the wettability between Al and SiC, making it a widely applied method [87,88,89].

At present, metals such as Ni-P [90], MgO [54], Cu [91], Cr [92] and Ti [93,94] are all deposited on the surface of SiCp as coatings to prevent contact between SiCp and molten Al and inhibit the occurrence of adverse interface reactions. Li et al. [95] the electroless plating method is adopted to Cu plating on the surface of SiCp, by means of pressureless infiltration technique was used to prepare the SiCp/Al composite materials, has not found the existence of Al_4_C_3_, explains the Cu coating can effectively inhibit really interface reaction, but it does not represent no interface reaction. When Rodriguez-Reyes et al. [96] studied SiC/Al-10Mg-12Si composites, although the presence of Al_4_C_3_ was not initially found, the powder behavior was still observed in the near interface region of the composites after the material was left for 4 months, indirectly confirming the existence of Al_4_C_3_.

Although the coating on the surface of SiCp can effectively prevent the excessive occurrence of interface reactions, this surface modification measure still has certain limitations in controlling the occurrence of slight interface reactions and hindering the mutual diffusion between atoms. It is difficult to generate a clean interface with neither reaction layer nor crystal layer in SiCp/Al composite materials. Some scholars began to use the method of molten alloy to control the slight interfacial reaction [97].

Wang et al. [98] successfully prepared β-SiCp/Al electronic packaging materials with different volume fractions using low-pressure infiltration. They investigated the influence of Mg and Si elements on the microstructure of the composites. The results showed that the addition of an appropriate amount of Mg in Al matrix induced an interface reaction between SiC and the oxide film of Mg, leading to the formation of magnesium aluminate (MgAl_2_O_4_), which improved the wettability at Al/SiC interface. The addition of Si to Al matrix inhibited the formation of the detrimental phase Al_4_C_3_ at the interface. The composites with a SiC volume fraction of 66% exhibited CTE and TC values that met the requirements for electronic packaging materials.

Apart from Si and Mg, Cu is also commonly added as an alloying element to improve the interface reaction between Al and SiC [99,100]. Fusheng P et al. [101] proposed that Cu, as a reactive element, reduces the reactivity of Al when it enriches at the interface, thus weakening the occurrence of Al/SiC interface reaction. However, Wang [102] showed that Cu does not enrich at Al-Cu/SiC interface. Yang and Zhang [103] fabricated SiC/Cu-Al composites with a high SiC volume fraction of 72.7% using low-pressure infiltration. The results demonstrated that the surface coating of SiC particles with a Cu layer effectively prevented the formation of Al_4_C_3_.

In addition, some scholars believe that the selection of reasonable preparation processes and parameters can control the interface reaction, Carvalho O et al. [104,105] studied the effects of different molding pressures and sintering temperatures on the interfacial reaction of SiCp/Al composites, and believed that effective control of the interfacial reaction of composites could be achieved by adjusting the sintering temperature and molding pressure and other process parameters of combined SiCp/Al composites.

Zhu Wanbo et al. [106] prepared SiC/Al composites with SiC content up to 65% by powder metallurgy, and studied the effect of heat preservation on the interface reaction under the condition of 630 °C and hot pressing pressure of 50 MPa, as shown in Table 3.

Currently, SiC/Al composites have been partially applied in the field of electronic packaging, but the research on interface reaction is still insufficient. In surface of SiCp plating, although coating can effectively inhibit undesirable interface reaction, also can make SiC enhancement effect is not fully play. To add Mg and Si elements such as Al matrix while it is possible to improve interface reaction and interface wettability, but also affect the thermal and mechanical properties of the Al. This is also the main reason why the thermal conductivity of SiC/Al composites is reduced, so people turn their attention to copper with higher thermal conductivity.

### 3.2. SiC/Cu Composite Materials

Taking inspiration from the successful development of SiC/Al electronic packaging materials, attention has shifted towards SiC-based composites with a Cu matrix (SiC/Cu). Compared to Al, Cu has various advantages, such as Cu’s electrical and thermal conductivity are better than Al, and the melting point is higher than Al. It is expected that SiC/Cu composites would exhibit higher TC, lower CTE, better mechanical strength, superior welding performance, and environmental stability. Additionally, with the abundance of microfabrication techniques, SiC/Cu composites are considered the best candidates for the next generation of electronic packaging materials [107,108,109]. SiC/Cu composites possess both the high electrical and TC of Cu and the strength and high-temperature stability of SiC, making them suitable for practical applications in electronic packaging, thermal management materials, and electrical contact materials [110,111].

Cu has a high TC, reaching 400 W/(m·K) [112]. In theory, SiC/Cu composites should exhibit better TC than SiC/Al composites. However, the interface reaction between SiC and Cu is challenging to control, leading to high interfacial thermal resistance, which hinders the full utilization of TC of the composites. Additionally, the wettability issue between SiC and Cu is also an important factor affecting TC. In this subsection, the research progress of SiC/Cu composites will be discussed in terms of interface reactions, wettability, and the influence of SiC content on the thermal and mechanical properties of the composites.

#### 3.2.1. Interface Reactions

SiC/Cu composites are highly favored for high-performance heat dissipation devices and electronic packaging applications due to their high TC and low CTE. Currently, most methods for fabricating SiC/Cu composites involve high-temperature and high-pressure conditions. However, the contact between SiC and Cu becomes unstable at high temperatures. Si dissolves into Cu, forming a Cu-Si solid solution, while pure carbon remains at the interface between Cu and SiC [113]. This interface reaction, represented by Equation (1) [114], significantly reduces TC of the composites. To prevent the interface reaction between Cu and SiC, diffusion barriers need to be established [115].
SiC + Cu→C + Cu_3_Si(1)

Guo et al. [116] studied the interface structure of SiCp/Cu composites and its influence on thermal conductivity, and compared the influence of different particle sizes of SiCp on thermal conductivity of composites. The microstructure of the SiC/Cu interface is shown in Figure 4, where a serious reaction occurs and the unreacted SiC particles are surrounded by a black Cu-Si layer. Figure 4e shows the results of Energy Dispersive Analysis of X-rays (EDAX) of the Cu-Si layer, in which the number of Cu atoms accounts for 91.8% and the number of Si atoms is 8.2%. The maximum number of atoms of solidly dissolved Si in Cu is about 9%. Combined with the results of selected area electron diffraction (SAED) of Cu-Si layer in Figure 4d, only the Cu phase is detected, it can be seen that Si element is solidly dissolved in Cu. In addition, as shown in Figure 4f, a polycrystalline graphite layer was found around the Cu-Si layer; As shown in Figure 4h, Si element is also found in the Cu matrix, and similar to the Cu-Si layer in the reaction zone, only the Cu phase is detected, as shown in Figure 4g. It can be seen from the above findings that Si element is sole-dissolved in Cu-Si layer, reaction zone and Cu matrix, resulting in Cu lattice distortion and affecting thermal conductivity. The effects of different particle sizes of SiCp on the thermal conductivity of composites are shown in Table 4, with the increase of particle size, the thermal conductivity of composites increases slightly, but it is far lower than the predicted value, for two main reasons: The first is that SiCp makes the copper crystal appear high density defect, which reduces the thermal conductivity of the copper matrix, and the second is that there is a serious interface reaction in the composite material.

Schubert et al. [117] applied a thin Molybdenum (Mo) film on the surface of SiCp using vapor deposition, which reacted with SiC to form molybdenum carbides and silicides, acting as an interface barrier layer. Liu Meng et al. [118] successfully prepared Mo coating on the surface of SiCp by magnetron sputtering method and SiCp/Cu composite by hot pressing sintering process, and studied the microstructure and interface structure of the composite as well as the influence of sputtering time on the thermal conductivity of the composite. Figure 5 shows the microstructure and interfacial structure of SiCp/Cu composites. Figure 5a, b show the microstructure of uncoated and Mo-coated SiCp/Cu composites respectively, indicating that the distribution of SiCp in SiCp/Cu composites is very uniform. However, the distribution of SiC particles in Mo-coated SiCp/Cu composites is more uniform, and there are almost no micropores and particle fractures. The interface structure of the SiCp/Cu composite without Mo coating is shown in Figure 5c, and the interface bonding between SiC and Cu is poor. The interface of the Mo coated SiCp/Cu composite is shown in Figure 5d, and the adhesion of the interface between SiC and Cu is significantly enhanced after Mo coating. Figure 5e,f show the backscatter micrographs of the interface structure of the uncoated and Mo-coated SiCp /Cu composites, respectively. Figure 5e shows that no other interface phases exist except Cu and SiC, while the thin white Mo interface layer can be clearly seen in the interface region of Figure 5f. It can effectively inhibit the harmful interfacial reaction and improve the interfacial wettability between SiC and Cu. The effect of sputtering time on the thermal conductivity of the composite is shown in Table 5, with the increase of magnetron sputtering time, the thermal conductivity of the composite material shows a trend of first increasing and then decreasing, and the main reason for the decrease in thermal conductivity is that with the extension of magnetron sputtering time, the thickness of Mo coating increases, but the thermal conductivity of Mo itself is smaller than that of Cu and SiC.

Xu et al. [119] fabricated Titanium(Ti)-coated SiCw/W-20Cu composites using hot-press sintering. Ti-coated SiCw, as shown in Figure 6, resulted in dense and uniformly structured SiCw/W-20Cu composites, partially suppressing the reaction between Cu and SiC.

Sundberg et al. [120] studied the effectiveness of four barrier layers in inhibiting the chemical reaction between Cu and SiCp. These four barrier layers were titanium nitride (TiN), diamond-like carbon, TiN/TiC/TiCN/TiN, and TiN/TiC/Al_2_O_3_. The results showed that the first two barrier layers, due to minor Si contamination in Cu, still caused a significant decrease in TC to 129 W/(m·K). On the other hand, the latter two barrier layers played a role in suppressing the interface reaction between Cu and SiCp to some extent, helping to reduce the TC loss in Cu.

Sundbert et al. [117] applied a Mo coating on the surface of SiCp using magnetron sputtering. The reaction between Mo and SiC formed Mo carbides and silicides, effectively preventing the interface reaction between Cu and SiC. SiCp/Cu electronic packaging materials were prepared using hot-press sintering. When the SiCp volume fraction was 40%, the TC of the composite reached 306 W/(m·K), and CTE at 30–100 °C was 11.2 × 10^−6^ K^−1^.

Similar studies have been conducted using NiP_2_ and Y_2_O_3_ [114], as well as coating materials such as Ti, Cr, Fe, Ni, and Al [121,122,123]. Although some coating layers have achieved certain effects, achieving perfect encapsulation of the surface of the particle reinforcement phase to completely eliminate direct contact reaction between Cu and SiC is challenging. Therefore, finding a comprehensive solution to the high-temperature reaction between Cu and SiC remains a major challenge in the development of Cu/SiC thermal sink materials.

#### 3.2.2. Wettability

The interfacial characteristics between the matrix and reinforcement are often described by the solid-liquid wettability when the liquid metal contacts the surface of the reinforcement [124]. After the droplets tend to stabilize, the contact angle θ is formed at the intersection of the liquid surface and the solid surface, also known as the wetting angle. 0 < θ < 90°, as shown in Figure 7a, is a solid-liquid system as an infiltration system, and when 90° < θ < 180°, as shown in Figure 7b, it is called a non-wetting system [125].

To enhance TC of the composite material, it is desirable to minimize the interfacial thermal resistance. However, the presence of porosity in the composite material significantly increases the interfacial thermal resistance, leading to a sharp decrease in TC [122]. Therefore, improving the densification of the composite material, which involves enhancing the wettability between the matrix and reinforcement, is a pathway to achieve excellent TC in electronic packaging materials.

Martinez et al. [126] demonstrated that at a temperature of 1100 °C under high vacuum conditions, the wetting angle between SiC and Cu was 140°, indicating poor wettability. Thus, improving the wettability between SiC and Cu is crucial for enhancing TC of SiC/Cu electronic packaging materials [127]. One commonly used technique to improve wettability is to introduce reactive elements into the matrix metal, which reduces the wetting angle and enhances the wettability between SiC and Cu matrix [14]. Active elements are usually metal elements, and the main active elements include Ti, Cr, etc. [128,129,130].

Zhang et al. [129] investigated the influence of the reactive element Ti on the wettability between SiC and Cu during the preparation of SiC/Cu composites via the low-pressure infiltration method using the sessile drop technique [131]. The results showed that the introduction of Ti as a reactive element at 1100 °C resulted in a wetting angle of only 15° between SiC and Cu. The segregation of Ti at the interface and the formation of TiC through the reaction between Ti and SiC significantly reduced the interfacial energy between SiC and Cu, thereby enhancing the wettability at the interface.

The method of adding active elements is mainly for the metal modified layer, and for the preparation of composite materials by fusion infiltration method. In addition, the method of adding active elements also has two unavoidable disadvantages: one is the influence of the residual active elements on the thermal conductivity of the matrix, and the other is the reaction or non-wetting phenomenon between the matrix and the reinforcement before the active elements are biased [128,132].

To improve the wettability between SiC and Cu, various methods such as chemical plating [133,134], sol-gel deposition [135], and electroplating [136] can be employed to deposit Cu or other coatings on the surface of SiC [137]. Hu et al. [138] successfully prepared high-performance SiC/Cu electronic packaging composites using a combination of chemical plating of Cu on SiC powder and high-speed flame spraying. The results showed that a uniform and dense Cu coating was obtained on the surface of SiC powder through chemical plating, enabling a high SiC volume fraction of over 55 vol% in the composite material. SiC/Cu composites fabricated using this method exhibited CTE and TC values that met the requirements for electronic packaging materials.

Chang Jiaqi et al. [139] studied the spreading of molten Cu on different substrates at 1250 °C, thus measuring the wettability of copper and SiC. The results show that the molten Cu is approximately ellipsoidal on the SiC ceramic substrate, the interface wetting Angle is 114.9°, and the wettability between Cu and SiC is very poor. Molten Cu is almost completely spread on pure nickel (Ni) substrate, the interface wetting Angle is 22.5°, Cu and Ni have excellent wettability. Molten Cu is almost completely spread on the SiC ceramic substrate after Ni plating, and the interface wetting Angle is 32.3°, indicating that Ni plating on the SiC surface can significantly improve the wettability of SiC and Cu.

Although electroless plating is a good choice, its efficiency is low, and palladium (Pd) and tin (Sn) in the activation and sensitization solution during the pretreatment of SiC surface will inevitably pollute the environment [140]. Lu Han et al. [141] deposited Cu on the surface of SiCp and SiCw by pulsed intermittent electrodeposition and prepared SiCp/Cu and SiCw/Cu composites combined with discharge plasma sintering. The mechanical properties and thermal conductivity of the composites are shown in Table 6, and the yield strength and hardness of SiC/Cu composites are higher than that of pure copper. The thermal conductivity of SiCw/Cu composites is lower than that of copper, and the yield strength, hardness and thermal conductivity of SiCw/Cu composites are higher than that of SiCp/Cu composites.

#### 3.2.3. SiC Content

The properties of copper matrix composites, such as hardness, thermal conductivity and thermal expansion coefficient, are closely related to reinforcement type, particle size and distribution [142,143]. Sorkhe et al. [144] prepared TiO_2_/Cu composites with different TiO_2_ mass percentages by mechanical alloying method to study the properties of the composites. Taha and Zawrah [145] studied the effects of different ZrO_2_ contents on the mechanical properties and electrical conductivity of Cu matrix. Akbarpour et al. [146] investigated the effects of different sizes of SiCp on the morphology, strength and wear behavior of Cu matrix. Fan et al. [147] prepared SiCp/Cu composites with different volume fractions by vacuum hot pressing method and found that the microhardness of the composites increased significantly as the volume fraction of SiCp increased.

The comprehensive properties of SiC/Cu composites are influenced by the SiC volume fraction, morphology, and interfacial reactions [148]. To investigate the fabrication process and formation mechanism of SiCw/Cu composites and further study the effect of SiCw content on the microstructure and mechanical properties of the composites, Cai et al. [111] fabricated SiCw/Cu composites with different SiCw mass fractions using vacuum hot pressing. They studied the influence of SiCw content on the mechanical properties of the composites. The results showed that with an increase in SiCw mass fraction, the hardness, porosity, and compression yield strength of the composites increased, while the tensile strength initially increased and then decreased.

A higher volume fraction of SiCw in Cu-based composites leads to a better reduction in CTE and an improvement in mechanical strength [128]. Due to the shape of whiskers and their proximity in the composite material, achieving a SiCw volume fraction exceeding 40% is challenging [149].

Yih and Chung [150] replaced the mixture of Cu powder and bare whiskers with Cu-coated SiCw and fabricated SiCw/Cu composites with SiCw volume fractions ranging from 33 to 54 vol% and a porosity of less than 5 vol%. The composites were then tested for their comprehensive properties. The results showed that the composites prepared using this method had low porosity, high hardness, high compressive yield strength, and low CTE. Among them, the 50 vol% SiCw/Cu composite exhibited the highest Vickers hardness of 260, while the 54 vol% SiCw/Cu composite had the lowest CTE of 9.6 × 10^−6^ °C (25–150 °C).

Zhang et al. [151] prepared SiCp/Cu composites with high volume fraction by electroless plating and powder metallurgy, and characterized the microstructure and thermal expansion coefficient of the composites. As shown in Figure 8 before and after SiCp electroless copper plating, the coating is small and compact. The effects of different volume content of SiCp on the properties of SiCp/Cu composites are shown in Figure 9. Figure 9a shows the effects of volume content of SiCp on the hardness and bending strength of the composites. As the content of SiCp increases, the bending strength of the composites decreases and the hardness increases. The thermal expansion coefficient of composites with different volume content of SiCp varies with temperature, as shown in Figure 9b. With the increase of SiC content, the thermal expansion coefficient of composites decreases.

Shan et al. [152] used hot pressing sintering technology to prepare β-SiCp/Cu electronic packaging materials with different volume fractions (10~60%) of β-SiCp. They conducted studies on the relative density, CTE, and TC of the composite material. The variation of TC and CTE with the volume fraction of β-SiCp in the β-SiCp/Cu electronic packaging composites is shown in Figure 10. As the volume fraction of β-SiCp increased, the relative density of β-SiCp/Cu composite gradually decreased, and both CTE and TC decreased accordingly.

Essam B. Moustafa et al. [153] prepared SiC/Cu nanocomposites with different mass fractions by hot pressing sintering, and studied the effects of different SiC contents on the properties of the composites at different temperatures. The effects of different SiC content on the properties of composites at different temperatures are shown in Figure 11. Figure 11a shows the effects of different SiC mass fraction and sintering temperature on the CET of nanocomposites. The CTE value decreases with the increase of SiC content and increases with the increase of sintering temperature. As shown in Figure 11b, the microhardness value of nanocomposites is greater than that of unreinforced Cu, and the microhardness increases with the increase of SiC content and sintering temperature. 

## 4. Summary and Outlook

Drawing lessons from the successful development of Al/SiC, Cu/SiC is considered the best option for the next generation of thermal management materials. However, despite years of exploration by researchers in the field of thermal management materials, significant success has not been achieved. The fundamental reason is the severe interface reaction between Cu matrix and SiC reinforcement material under high-temperature conditions, leading to a significant decrease in TC of the composite material.

Based on the current research status of electronic packaging materials, MMCs have significant advantages in terms of performance and cost. Therefore, future research on metal-based electronic packaging materials can focus on the following two aspects: Replacing traditional composite fabrication methods with low-temperature and low-pressure preparation methods to prevent interface reactions from occurring at the source; Selecting superior reinforcement materials and developing MMCs with better thermal and physical properties for electronic packaging applications.

## Figures and Tables

**Figure 1 micromachines-14-01491-f001:**
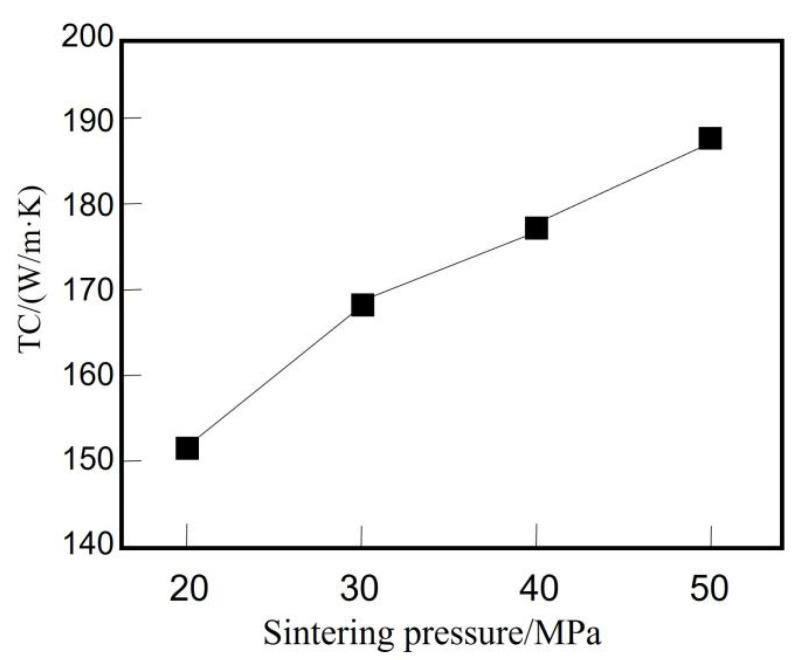
Relationship of TC of the SiCp/Al composites and sintering pressure [58].

**Figure 2 micromachines-14-01491-f002:**
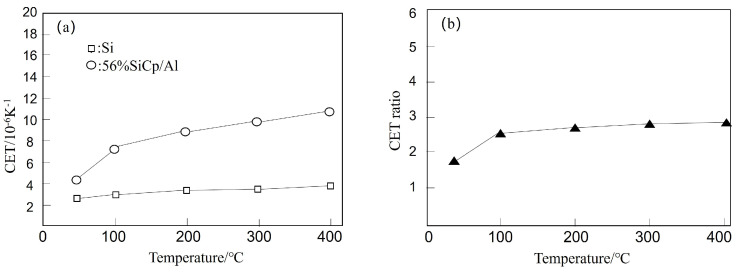
Temperature-dependent variations of CTE of 56% SiCp/Al electronic packaging material and Si chip material (**a**), and their ratio (**b**) [60].

**Figure 3 micromachines-14-01491-f003:**
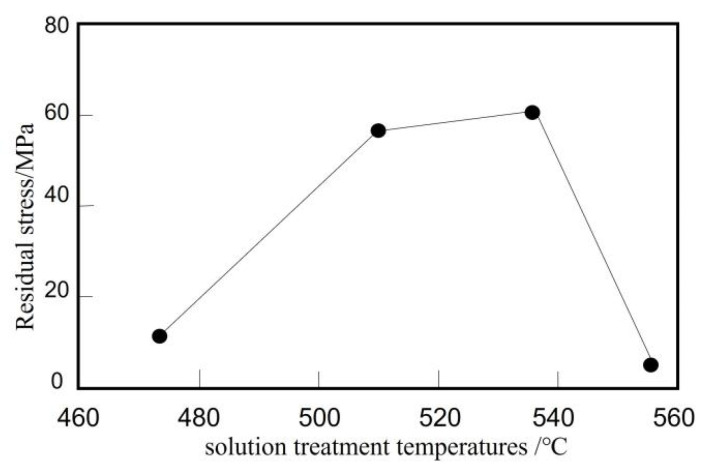
Residual stress in the aluminum matrix composite based on different solution treatment temperatures [71].

**Figure 4 micromachines-14-01491-f004:**
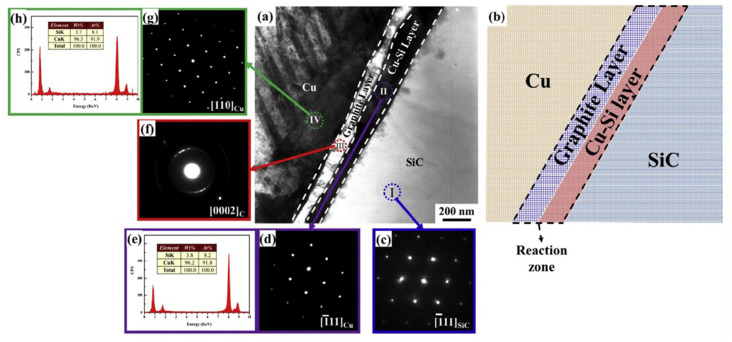
Interfacial structure of SiCp/Cu composite. (**a**) TEM image; (**b**) schematic diagram of (**a**); (**c**) SAED of I area; (**d**) SAED of II area; (**e**) EDAX of II area; (**f**) SAED of III area; (**g**) SAED of IV area; (**h**) EDAX of IV area [116].

**Figure 5 micromachines-14-01491-f005:**
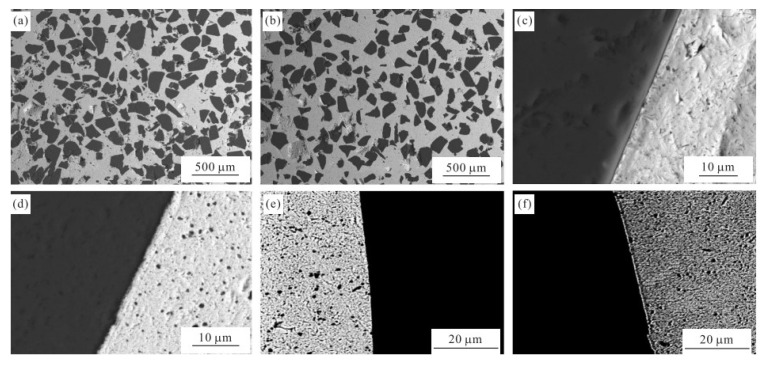
Micrographs of the SiCp /Cu composites: (**a**) Uncoated SiCp/Cu; (**b**) 9 h Mo-coated SiCp/Cu; (**c**) Interface structure of Uncoated SiCp/Cu; (**d**) Interface structure of Mo-coated SiCp/Cu; (**e**) the back scattered micrograph of uncoated SiCp/Cu composites; (**f**) the back scattered micrograph of Mo-coated SiCp /Cu composites [118].

**Figure 6 micromachines-14-01491-f006:**
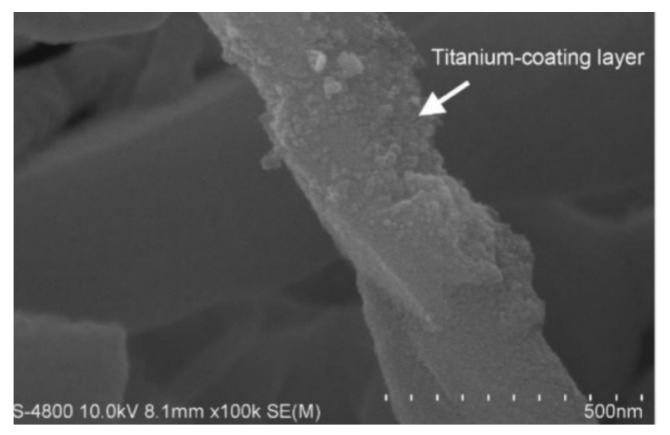
Preparation of Ti-plated SiCw using vacuum slow gas phase deposition process [119].

**Figure 7 micromachines-14-01491-f007:**
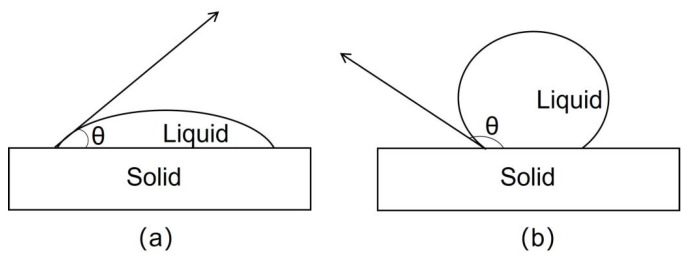
Droplet shape and contact angle in solid-liquid system; (**a**) Wetting system; (**b**) Non-wetting system [125].

**Figure 8 micromachines-14-01491-f008:**
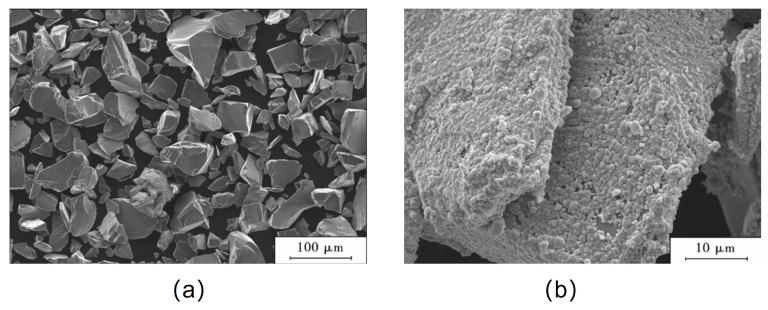
The microstructure of SiC particles before and after chemical plating; (**a**) Before chemical plating; (**b**) After chemical plating [151].

**Figure 9 micromachines-14-01491-f009:**
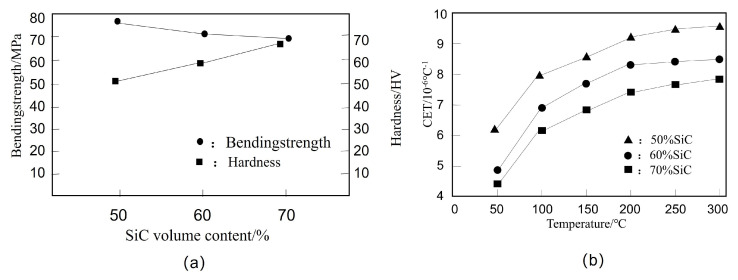
The effect of the volume content of SiCp on the properties of SiCp/Cu composites: (**a**) Effect of SiC volume content on flexural properties and hardness of SiC/Cu composites; (**b**) The coefficient of thermal expansion of the composite material with temperature under different SiC volume fractio [151].

**Figure 10 micromachines-14-01491-f010:**
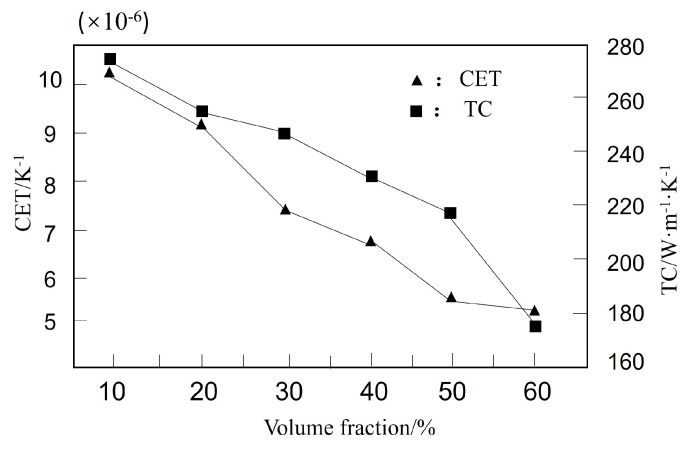
CTE and TC curves of β-SiCp/Cu-volume fraction [152].

**Figure 11 micromachines-14-01491-f011:**
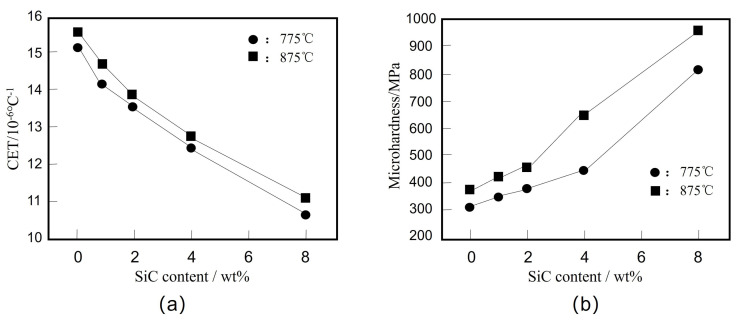
Effect of SiC content on properties of composites at different temperatures: (**a**) Effects of SiC weight percentage and sintering temperature on CTE values of nanocomposites samples; (**b**) Microhardness of the Cu/SiC nanocomposite specimens [153].

**Table 1 micromachines-14-01491-t001:** Thermal conductivity of composites as a function of pressing height [58].

Pressing Height/cm	TC/(W/m·K)
40	170
60	179
80	175.9

**Table 2 micromachines-14-01491-t002:** Relationship between SiC volume fraction and performance of composite materials [60].

SiC Volume Fraction (%)	TC (W/m·K)	CTE (10^−6^ K^−1^)	Compressive Strength (MPa)	Flexural Strength (MPa)	Experimental Density (g/cm^3)^	Density
40	164	12.60	350.1	347.5	2.89	0.997
56	176	9.25	496.2	401.6	2.94	0.987
63	178	8.21	685.1	437.0	2.97	0.985

**Table 3 micromachines-14-01491-t003:** Effect of holding time on composites [106].

Holding Time/h	Interface Reaction	TC/(W/m·K)	CET/(10^−6^/°C)
6	NO	211	7.5
12	Mild reaction, with MgAl_2_O_4_ production	216	7.3

**Table 4 micromachines-14-01491-t004:** Effect of different particle sizes on thermal conductivity of composites [116].

Materials	Volume Amount/vol.%	Average Particles Size (μm)	TC/(W/m·K)
SiCp/Cu	55	10	69.6 ± 1.2
	55	20	73.1 ± 2.5
	55	63	78.5 ± 1.9

**Table 5 micromachines-14-01491-t005:** Thermal conductivity of SiCp/Cu composites under different magnetron sputtering times [118].

Sputtering Time/h	Volume Fraction/%	TC/(W/m·K)
0	50.576	165.560
3	51.370	224.160
6	49.825	251.528
9	51.369	274.056
12	49.110	265.501

**Table 6 micromachines-14-01491-t006:** Mechanical and thermal conductivity of sintered Cu, SiCp/Cu and SiCw/Cu [141].

Materials	Yield Strength/MPa	Microhardness/HV	TC/(W/m·K)
Cu	162 ± 3.3	80 ± 1.2	324.6
SiCp/Cu	306 ± 3.3	126 ± 1.1	246
SiCw/Cu	346 ± 3.6	137 ± 1	261

## Data Availability

Data availability is not applicable to this article as no new data were created or analyzed in this study.

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
