# Peer review of "Advancements in SiC-Reinforced Metal Matrix Composites for High-Performance Electronic Packaging: A Review of Thermo-Mechanical Properties and Future Trends"

_micromachines, 2023, doi:10.3390/mi14081491_

Round 1

Reviewer 1 Report

his paper provides a comprehensive overview of the research progress in silicon carbide metal based electronic packaging materials. The paper has clear and logical writing, making it suitable for publication in this journal. Here, there are some issues need to be addressed, then we can further consider it for publication or not.

1. Some language expressions need to be further polished, accurate, vivid and concise language should be used as far as possible, and the layout format needs to be further planned.

2. In the preparation methods of SiC/Al composites, the authors should point out the main reasons for the limitations of traditional manufacturing processes and costs.

3. The examples of Mou Zonghua and Essam B. Moustafa do not correspond to the classification or subheading.

4. The thermal conductivity units in the table should be bracketed to avoid ambiguity.

5. The lack of research on interface regulation of SiC/Al composites has not been pointed out.

Some language expressions need to be further polished, accurate, vivid and concise language should be used as far as possible, and the layout format needs to be further planned.

Author Response

Authors’ Responses to Review Comments

Dear Micromachines Editor,

Re: " Advancements in SiC-Reinforced Metal Matrix Composites for High-Performance Electronic Packaging: A Review of Thermo-Mechanical Properties and Future Trends" by Bing Niu, Liyan Lai, Yuxiao Bi, Yigui Li, and Zhuoqing Yang, Article reference: micromachines-2507048

To begin with, we would like to express our thanks to the three reviewers for their careful review and constructive suggestions regarding our manuscript. After getting the reviewers’ comments, we studied it carefully and tried our best to revise and improve the manuscript according to the comments. We gave point-by-point responses to the reviewers’ comments, as detailed below. In the revised manuscript, all textual revisions are marked in red. A clean final version of our revised manuscript is uploaded as well.

Thanks for all the help.

Yours sincerely

Bing Niu

Response to the reviewer #1:

The authors wish to thank the anonymous reviewer for his/her critical comments that helped immensely in improving this manuscript. The changes have been made in this manuscript in red fonts, and the detailed explanations are as follows:

Reviewer #1 Evaluations: 

Overall Rating: Reconsider based on responses to issues raised by reviewers 

Reviewer #1 (Significance Level Comment): 

His paper provides a comprehensive overview of the research progress in silicon carbide metal based electronic packaging materials. The paper has clear and logical writing, making it suitable for publication in this journal. Here, there are some issues need to be addressed, then we can further consider it for publication or not. 

Reviewer #1 (Remarks): 

  1. Some language expressions need to be further polished, accurate, vivid and concise language should be used as far as possible, and the layout format needs to be further planned. 

Response:

We appreciate the reviewer’s comment. We have made comprehensive revisions to the logic and word accuracy of the language expression in the manuscript, and the overall layout has also been adjusted to make the article look more beautiful.

  1. Line 145: In the preparation methods of SiC/Al composites, the authors should point out the main reasons for the limitations of traditional manufacturing processes and costs.

Response:

Thank you very much for the careful review. In our manuscript, we pointed out that the preparation of high-volume fraction of SiC/Al composite materials is the main reason for the limit cost and technique. The related expression has been added in the revised manuscript as'--- Due to the limitations of cost and process, the traditional powder metallurgy method and stir casting method are difficult to be applied to the manufacture of high-volume fraction SiC/Al composite.'  (Page 4, Line 145-147)

  1. Line 218: The examples of Mou Zonghua and Essam B. Moustafa do not correspond to the classification or subheading.

Response:

We sincerely apologize for the mistake. The example of Mou Zonghua in the manuscript describes that a suitable position has been spared and a subtitle has been modified.  (Page 6, Line 218-225 and Line 556)

  1. The thermal conductivity units in the table should be bracketed to avoid ambiguity.

Response:

Thank you for the careful review. The description of thermal conductivity units in the manuscript does lead to ambiguity, and we have placed thermal conductivity units in brackets in the manuscript to avoid misunderstanding.    (Table 1-6)

  1. Line 351: The lack of research on interface regulation of SiC/Al composites has not been pointed out.

Response:

Thank you for the careful review and in-depth consideration of this issue. In this manuscript, we summarize the shortcomings of the existing interface regulation methods for SiC/Al composites. The related expression has been added in the revised manuscript as '--- In surface of SiCp plating, although coating can effectively inhibit undesirable inter-face reaction, also can make SiC enhancement effect is not fully play. To add Mg and Si elements such as Al matrix while it is possible to improve interface reaction and inter-face wettability, but also affect the thermal and mechanical properties of the Al. This is also the main reason why the thermal conductivity of SiC/Al composites is reduced, so people turn their attention to copper with higher thermal conductivity.'  (Page 8, Line 351-357)

Once again, we thank the referees very much for the careful review and the constructive comments and suggestions.

Reviewer 2 Report

This paper provides a comprehensive review of the current research progress in SiC-reinforced aluminum (Al)-based and copper (Cu)-based electronic packaging materials, focusing on the factors influencing their thermal and mechanical properties and the measures taken to enhance them. The study holds significant academic value, with a paper format that aligns well with the journal's requirements. The text is clear, logically organized, and highly recommended for publication in this journal.

However, there are a few issues that need to be addressed before further consideration can be given to its publication. Once these concerns have been resolved, we can proceed with the evaluation process.

1) After the introduction of SiCw, no corresponding examples are given to supplement the explanation.

2)The description of the main causes of residual stress in SiC/Al composites is repeated.

3)In the finite element simulation study of residual stress of SiC/Al composites, some examples reviewed do not belong to the finite element simulation study.

4)Minor interface reactions and severe interface reactions of SiC/Al composites are not given in detail.

5)The advantages of Cu in electronic packaging compared with Al are not explained.

6)An example of SiC mass fraction appears in the review of the effect of SiC volume fraction on the properties of SiC/Cu composites.

English writing is fine, only few typos have to be corrected.

Author Response

Authors’ Responses to Review Comments

Dear Micromachines Editor,

Re: " Advancements in SiC-Reinforced Metal Matrix Composites for High-Performance Electronic Packaging: A Review of Thermo-Mechanical Properties and Future Trends" by Bing Niu, Liyan Lai, Yuxiao Bi, Yigui Li, and Zhuoqing Yang, Article reference: micromachines-2507048

To begin with, we would like to express our thanks to the three reviewers for their careful review and constructive suggestions regarding our manuscript. After getting the reviewers’ comments, we studied it carefully and tried our best to revise and improve the manuscript according to the comments. We gave point-by-point responses to the reviewers’ comments, as detailed below. In the revised manuscript, all textual revisions are marked in red. A clean final version of our revised manuscript is uploaded as well.

Thanks for all the help.

Yours sincerely

Bing Niu

Response to the reviewer #2:

The authors wish to thank the anonymous reviewer for his/her critical comments that helped immensely in improving this manuscript. The changes have been made in this manuscript in red fonts, and the detailed explanations are as follows:

Reviewer #2 Evaluations: 

Overall Rating: Reconsider based on responses to issues raised by reviewers 

Reviewer #2 (Significance Level Comment): 

This paper provides a comprehensive review of the current research progress in SiC-reinforced aluminum (Al)-based and copper (Cu)-based electronic packaging materials, focusing on the factors influencing their thermal and mechanical properties and the measures taken to enhance them. The study holds significant academic value, with a paper format that aligns well with the journal's requirements. The text is clear, logically organized, and highly recommended for publication in this journal.

However, there are a few issues that need to be addressed before further consideration can be given to its publication. Once these concerns have been resolved, we can proceed with the evaluation process. 

Reviewer #2 (Remarks): 

  1. Line 117: After the introduction of SiCw, no corresponding examples are given to supplement the explanation. 

Response:

Thank you very much for the careful review. Examples of SiCw reinforced metals have been added later. The related expression has been added in the revised manuscript as '---SiCw is widely used as reinforcement. Xu Zhang et al. prepared SiCw/Al composites by hot isostatic pressing process. Tayebi M et al. prepared SiCw/Mg composites by stir casting. Jiang Feng et al. prepared SiCw/Cu composites by powder metallurgy and hot extrusion. Liyan Lai et al. prepared SiCw/Ni composites with excellent mechanical properties by electrodeposition.'  (Page 3, Line 117)

  1. Line 212: The description of the main causes of residual stress in SiC/Al composites is repeated. 

Response:

Thank you very much for the careful review. The repetitions describing the main causes of residual stress in SiC/Al composites have been removed.  (Page 5, Line 212)

  1. Line 218: In the finite element simulation study of residual stress of SiC/Al composites, some examples reviewed do not belong to the finite element simulation study.

Response:

Thank you for the meticulous review. Sorry for the misclassification of this example. The nonconforming examples have been moved out and put in place.  (Page 6, Line 218)

  1. Line 288: Minor interface reactions and severe interface reactions of SiC/Al composites are not given in detail.

Response:

Thank you for the careful review. Typos have been corrected in the manuscript. The severe interfacial reactions and the mild interfacial reactions of SiC/Al composites have been described in detail. The related expression has been added in the revised manuscript as  '---Adding appropriate amount of magnesium into the aluminum matrix, Silicon dioxide (SiO2) on the surface of SiCp can have a slight interfacial reaction with magnesium, forming MgAl2O4 between the interfacial phases, improving the wetness of the interface phase of the composite, which is conducive to improving the performance of the composite. In the process of SiC/Al composites prepared at high temperature and pressure, serious interfacial reactions will occur, and by-product aluminium carbide (Al4C3) will be generated, and Al4C3 will become the thermal diffusion barrier layer at the interface, which will greatly reduce the thermal conductivity of the composite.'   (Page 7, Line 288-295)

  1. Line 361: The advantages of Cu in electronic packaging compared with Al are not explained.

Response:

Thank you for your reminder, In the revised manuscript, we added the advantages of copper as a packaging material, and compared it with aluminum in specific properties. The related expression has been added in the revised manuscript as '--- Such as Cu's electrical and thermal conductivity are better than Al, and the melting point is higher than Al.'  (Page 9, Line 361)

  1. Line 68: An example of SiC mass fraction appears in the review of the effect of SiC volume fraction on the properties of SiC/Cu composites. 

Response:

Thank you for your comment. We have changed the original subtitle from "SiC volume fraction" to "SiC content" in the revised manuscript.   (Page 13, Line 556)

Once again, we thank the referees very much for the careful review and the constructive comments and suggestions.
